# Personality Assessment Based on Electroencephalography Signals during Hazard Recognition

**Mohan Wang and Pin-Chao Liao \***

Department of Construction Management, Tsinghua University, Beijing 100084, China
\* Correspondence: pinchao@tsinghua.edu.cn

**Abstract:** Hazard recognition assisted by human–machine collaboration (HMC) techniques can facilitate high productivity. Human–machine collaboration techniques promote safer working processes by reducing the interaction between humans and machines. Nevertheless, current HMC techniques acquire human characteristics through manual inputs to provide customized information, thereby increasing the need for an interactive interface. Herein, we propose an implicit electroencephalography (EEG)-based measurement system to automatically assess worker personalities, underpinning the development of human–machine collaboration techniques. Assuming that personality influences hazard recognition, we recorded the electroencephalography signals of construction workers and subsequently proposed a supervised machine-learning algorithm to extract multichannel event-related potentials to develop a model for personality assessment. The analyses showed that (1) the electroencephalography-assessed results had a strong correlation with the self-reported results; (2) the model achieved good external validity for hazard recognition-related personality and out-of-sample reliability; and (3) personality showed stronger engagement levels and correlations with task performance than work experience. Theoretically, this study demonstrates the feasibility of assessing worker characteristics using electroencephalography signals during hazard recognition. In practice, the personality assessment model can provide a parametric basis for intelligent devices in human–machine collaboration.

**Keywords:** hazard recognition; Big Five traits; electroencephalogram; machine learning; assessment

## 1. Introduction

Serious accidents and increased casualty rates in the construction industry are an increasing concern. In the United States, less than 5% of the workforce is employed in construction; however, the industry accounts for one in five work-related deaths (OSHA, 2018), many of which are caused by human factors (Beus et al., 2015) [1]. The system safety theory suggests that human recognition ability is limited; therefore, the human perception of hazards is likely to be biased or even incorrect (Senders et al., 1985 [2]; Woodcock, 2014 [3]). This is a significant reason why human factors contribute to accidents. Current research aims to improve the effectiveness of hazard identification by utilizing machines to assist in the process of human hazard recognition (Martinez-Marquez et al., 2021 [4]) as the first step toward the prevention of accidents. However, the process of developing a human–machine collaboration (HMC) is complex and requires matching individual characteristics to improve its effectiveness (Jeelani et al., 2017 [5]). Studies have revealed that HMC is more expensive because of the limited understanding of the psychological mechanisms of engineering hazards (Namian et al., 2018 [6]). Therefore, it is necessary to develop an efficient machine-driven hazard identification process. This can be achieved by integrating human personality, behavior, and cognitive factors into a machine (Villani et al., 2018 [7]).

Studies have assessed the influence of worker personalities (Hasanzadeh et al., 2019 [8]), safety risk perception (Namian et al., 2016 [9]; Pandit et al., 2019 [10]), behavior (Jin et al., 2019 [11]), knowledge (Namian et al., 2016 [9]), and attention (Jeelani et al., 2018 [12]) on

the hazard recognition process. Personality is defined as a collection of behavioral, cognitive, and affective models that have evolved from biological and environmental factors (Philip & Gerald, 2009 [13]). As an essential psychological trait that affects human perception (Gao et al., 2020 [14]), attention (Jeelani et al., 2018 [12]), and other factors, personality exerts an important influence on an individual's hazard recognition ability. However, many studies have relied on questionnaires (De Schutter, 2021 [15]), social networking platforms (Bhardwaj et al., 2016 [16]; Connelly & Ones, 2010 [17]), and surveys to obtain information on personality. These methods increase labor and time costs and carry the potential for falsification. Therefore, recent research has identified the intuitive (physiologically driven) HMC technique as a key area for future research (Villani et al., 2018 [7]).

Our study aimed to identify personality traits related to an individual's hazard recognition process by extracting physiological signals using electroencephalography (EEG). Furthermore, we attempted to develop an assessment model for worker personalities based on brain activity during a hazard recognition task.

## 1.1. Literature Review

### 1.1.1. Personality Traits Influence Hazard Recognition Performance

Prior studies have shown that personality changes (and other psychological traits) affect an individual's responses to hazardous situations. Moreover, personality traits are considered individual characteristics that influence hazard identification (Gao et al., 2020 [14]). Several relationships between personality traits and safety performance have been identified. With regard to agreeableness, it has been suggested that a higher level of agreeableness is linked to a greater propensity for teamwork, which is more conducive to generating positive safety perceptions (Gao et al., 2020 [14]). Individuals with lower agreeableness are more aggressive, and thus, more likely to be involved in hazardous situations (Templer, 2012 [18]). Nonetheless, because agreeableness is primarily related to teamwork (Mount et al., 1998 [19]), its relevance to features related to hazard identification, such as attention, is relatively weak (Hasanzadeh et al., 2019 [8]). The positive correlation between conscientiousness and the frequency of workers shifting their attention to dangerous situations suggests that highly conscientious workers may be more aware of danger and are better able to identify it than their less conscientious counterparts (Hasanzadeh et al., 2019 [8]). This also indicates a strong correlation between conscientiousness and hazard recognition. Individuals with high neuroticism are more likely to be distracted, negative, and stressed, whereas individuals with low neuroticism tend to be calmer and more relaxed (Costa Jr et al., 1986 [20]). Neuroticism influences the attention of workers and is one of the multiple factors that affect hazard recognition performance. Currently, no consensus exists regarding the correlation between extraversion and safety. Most studies have suggested that individuals with high extraversion are more susceptible to external stimuli than their counterparts (Barrick et al., 2013 [21]; Christian et al., 2009 [22]; Jonah, 1997 [23]). Furthermore, extraverts perform less effectively on tasks that require vigilance (Koelega, 1992 [24]), and they may conduct tasks with less effort than introverts, suggesting that extraverts are more likely to engage in hazardous situations and have a lower level of hazard recognition. However, this also suggests that individuals with high extraversion often ask for more instructions when determining safety risks, which increases their awareness of their situation (Henning et al., 2009 [25]), thereby making them more likely to recognize hazards. The relationship between openness and hazard recognition performance is predominantly reflected in the fact that open individuals may focus on scenes to gain information from hazardous stimuli (Costa Jr et al., 1986 [20]), resulting in better hazard recognition performance.

### 1.1.2. Limitations of Self-Reporting for Personality

The Big Five trait theory is a well-known classification of personality traits (Rothmann & Coetzer, 2003 [26]). Initially developed in the 1990s, this theory identifies five factors, typically referred to as openness, conscientiousness, extraversion, agreeableness,

and neuroticism (Costa & McCrae, 1992 [27]). Many current statistics on the Big Five traits rely on the statistical scales of these traits, testing an individual by the conformity of choice and relevant descriptions (Plaisant et al., 2010 [28]). The questionnaire developed for the scale is widely accepted in the psychology community (Gosling et al., 2003 [29]). Accordingly, social networking platforms solicit and assess the personalities of individuals using this tool (Bhardwaj et al., 2022 [30]). However, because the personality questionnaire results derived from the scale rely on the participant's self-reported values, they can easily be falsified (Viswesvaran & Ones, 1999 [31]). For example, a worker may deliberately embellish his/her answers in areas, such as responsibility and concentration, when answering a personality questionnaire to improve their personality score and make it easier for them to get a job. This for-profit embellishment makes it difficult to obtain true personality data in practice. Therefore, research has called for alternative methods to replace self-reported personality measures (Morgeson et al., 2007 [32]), including indirect measures of participants' personalities by observing their behavioral models (Gawronski & Houwer, 2014 [33]). Accordingly, herein, we attempted to develop a system to automatically assess worker personalities during hazard recognition tasks.

### 1.1.3. Refining Individual Characteristics Based on EEG Signals

An EEG captures brain wave models by recording electrical activity on the scalp, which is convenient but time-consuming to measure (Chatterjee et al., 2013 [34]), and provides information on an individual's brain activity characteristics and psychophysiological states (Sulavko et al., 2020 [35]). Previous research has also shown that all Big Five personality traits are linked to emotional responses (Letzring & Adamcik, 2015 [36]) and can influence an individual's emotional experience; for example, extraverts are more likely to experience positive emotions than introverts (John et al., 2008 [37]). Further studies of event-related potentials (ERPs) have suggested that personality influences an individual's neural response to emotional stimuli (Lou et al., 2016 [38]). EEG signals have also been used to extensively study the extraction of human emotions (Miranda-Correa, 2021 [39]; Tian, 2021 [40]), behaviors (Jin et al., 2021 [41]), and cognition (Landau et al., 2020 [42]; Rogala et al., 2021 [43]). The stable relationship between personality and EEG signals ensures the feasibility of personality inference based on brain activity (Zhao et al., 2018 [44]).

Recent relevant studies have relied on the EEG and other methods to assess personality traits. For example, Baumgartl et al., (2020 [45]) accurately identified extraverts with a performance rating accuracy of 67% and achieved a balanced accuracy of 60.6% with resting-state EEG data. Subramanian et al., (2018 [46]) used an EEG to perform a binary evaluation of emotions and personality while Zhao et al., (2018 [44]) classified the Big Five traits of participants by capturing their EEG signals during material viewing. Therefore, the evaluation of EEG signals is a viable method to assess and predict personality traits. However, these studies commonly provided binary judgments, such as high or low levels of personality, which are less applicable than the refined judgments presented by the scores obtained from the original questionnaire.

Many studies have been conducted on people from the construction industry to monitor their status at construction sites using EEG data. Wang et al., (2019 [47]) used wavelet packet decomposition to process EEG data and measure construction workers' risk perception based on their level of alertness, and subsequently provide quantitative indicators of vigilance. Jebelli et al., (2018 [48]) took a fixed-window approach and used a Gaussian support vector machine (SVM) to classify the collected EEG signals, generating a model that could be used to detect stress in workers. Ke et al., (2021 [49]) used EEG data to investigate the correlation between distraction and brain activity, providing a method for the objective monitoring of worker distraction. Some studies have also focused on hazard identification. Noghabaei et al., (2021 [50]) combined data from EEGs and eye tracking in an immersive virtual environment and classified it using a Gaussian SVM to predict when a safety hazard could be successfully identified. Chen et al., (2022 [51]) extended the results of previous studies on perceptual decision-making by establishing a scenario

experiment simulated with high realism. We have summarized these studies in Table 1. This also provided an explanation for the possible existence of different neural-influencing mechanisms in individual workers through intrinsic factors of risk propensity and extrinsic factors of injury experience. In terms of research participants, the aforementioned studies focused on the short-term states of workers, such as vigilance, stress, distraction, and hazard recognition time in construction sites, and considered personality as an essential attribute of workers that has developed over time. In addition, the continuous personality assessment values also showed the same form as the scores obtained from the manual questionnaire collection, thereby allowing for the resolution of the binary problem in automated personality collection.

**Table 1.** Comparison with previous studies.

| Related Works | Methods Used in the Related Work | Methods Used in the Present Study |
| --- | --- | --- |
| Wang et al., (2019) [47] | A total of 30 potential indicators at three risk levels; three optimal indicators. | A number of models were obtained using regression in supervised machine learning. |
| Ke et al., (2021) [49] | The voltage, time–frequency magnitude, and indicators of frequency bands were calculated; SVM. | A nested model containing inner and outer loops was established in which the inner loop established a sparse regression model and the outer loop set a lockbox. The optimal regression model was found by changing the *p*-value threshold set by the inner loop. |
| Jebelli et al., (2018) and Noghabaei et al., (2021) [48] | The obtained electroencephalography data were classified, and the highest classification accuracy with a Gaussian SVM was obtained. | |

SVM, support vector machine.

## 2. Methodology

### 2.1. Overview

In the experiment, we recorded the EEG signals of workers during construction site image observation. Thereafter, we extracted the ERPs and used a method that included inner and outer loops to train the assessment model to completion (Figure 1). The model properties were subsequently evaluated.

1. The correlation between the EEG-assessed and actual values of a worker's personality was analyzed to assess the assessment properties of the model.
2. The EEG-assessed results were used to assess other participant characteristics and verify the external validity of the model.
3. The model was applied to a lockbox to evaluate the out-of-sample reliability.
4. Several types of data were extracted to train the assessment model for further validation.

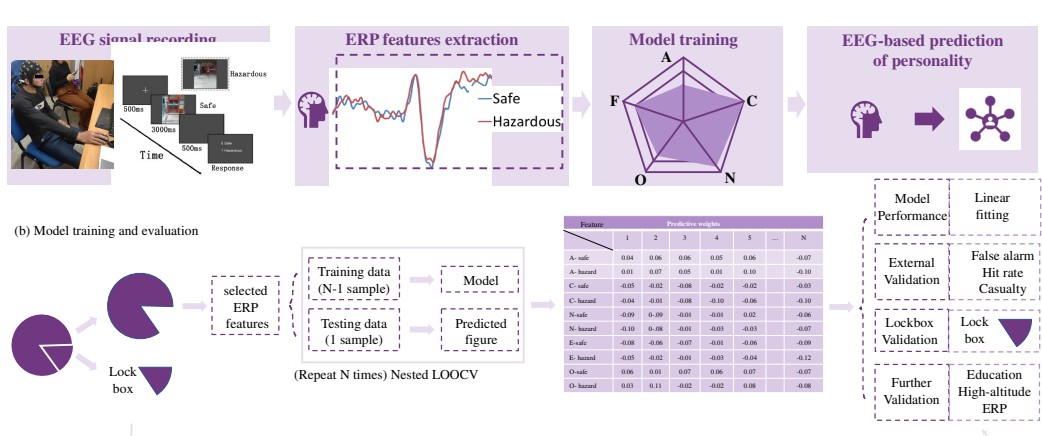

**Figure 1.** Experimental flowchart. (**a**) EEG data recording; (**b**) model construction. EEG, electroencephalography and ERP, event-related potential.

*2.2. Participants and Materials*

A total of 85 workers from the construction site of an office building in Beijing participated in the experiment. From them, 24 workers were excluded because of issues with data recording, physical and experiential factors, contradictory responses, dissatisfaction with the recorded EEG signals, or extracted ERP features. Finally, 61 male participants with an average age of 41.1 years (range: 21–60 years) were included in the study. As image stimuli, we selected 60 safety and 60 hazard images captured at actual construction sites. The images contained various construction scenes and hazard types that would typically cause solid visual stimuli. The safety conditions were recorded using images after hazard rectification (Figure 2). All images were obtained from a previously developed database (Xu et al., 2019 [52]). Information on the participants' personalities was obtained through a questionnaire in which the participants completed a 7-point scale of behavioral and psychological traits in everyday life (the questionnaire was illustrated in the form of a cartoon villain, and all images were easy to understand in everyday life). A score of 1 indicated extremely unlikely, while a score of 7 indicated extremely likely.

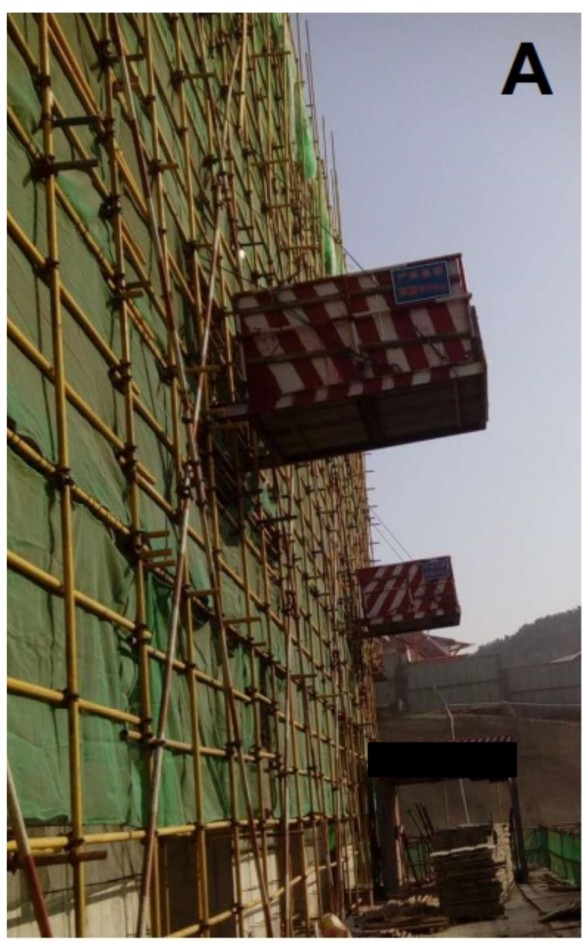 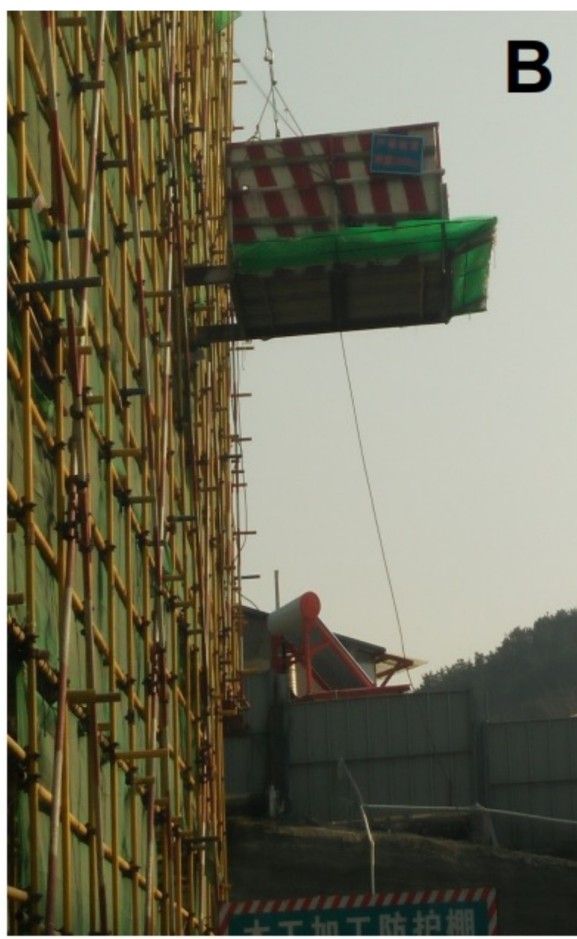

**Figure 2.** A pair of sample stimuli in the experiment. (**A**) Hazardous: There is no safety net at the outer edge of the unloading steel platform. (**B**) Safe: After rectification, the outer edge of the unloading steel platform is equipped with a safety net.

*2.3. Experimental Procedure*

The experiment was conducted in the office of scholars at the Tsinghua University. The participants completed a questionnaire before the experiment began. The main experiment consisted of 60 groups of hazardous and safe images and followed a specific procedure, which has been previously described (Wang et al., 2022 [53]).

### 2.3.1. EEG Signal and Preprocessing

We used a portable wireless EEG signal amplifier (NeuSen.W32, Neuracle, China) to record the EEG data with a sampling frequency of 250 Hz. The electrodes were placed according to the international 10–20 system, and the signals were recorded continuously on 30 EEG channels. The EEG signal used CPz as the reference electrode and FPz as the ground electrode. During the experiment, the resistance values of all electrodes were maintained below 10 kOhm. The raw EEG data were processed using a bandpass filter (1–30 Hz). An independent component (IC) analysis was applied to the continuous EEG data to eliminate artifacts resulting from muscle movements, eye movements, and other noise. For every participant, we removed an average of one to three ICs associated with artifacts. The ICs were then back-projected onto each channel to reconstruct high-quality data. The trials were cut to 1200 ms (200 ms before stimulation to 1000 ms after stimulation), and all trials with potential amplitudes exceeding $\pm 100$ μV were eliminated to avoid the possible influence of artifacts. After processing, nine participants were excluded due to a trial retention of less than 60%. The safe and hazardous data of each remaining participant were averaged and calibrated by the baseline.

### 2.3.2. Feature Selection and Model Training

The experiment focused on components related to the visual task, which included P1, N2, and P3 (120–180, 180–240, and 260–340 ms, respectively). We performed bivariate Spearman correlations between the average amplitudes of the three components for the safe and hazardous stimuli and calculated the Big Five trait questionnaire-based results to test the feasibility of using ERPs to initially assess the Big Five traits. All channels within each region of interest (ROI) were averaged to test the relationship between the mean ERP amplitudes of each ROI and personality.

For the assessment of personality, the regression model was trained using full-channel potential values at the total sampling points averaged for each of the two types of stimuli: safe and hazardous. Generally, the full-channel ERP data provided detailed spatial and temporal information to assess personality traits. The average multichannel ERP across the two types of stimuli ultimately contained 2 (image type) $\times$ 30 (EEG channels) $\times$ 300 (sampling points) = 18,000 features (per participant). Nested cross-validation was used to ensure the reliability of the model as the dimension of the feature was much larger than the size of the sample (Cu et al., 2016 [54]). The cross-validation model contained two loops: inner and outer. The inner model comprised a sparse regression model to select traits with potentially significant contributions to the dependent variable (i.e., personality). Next, the selected features were dimensionalized, and a *p*-value threshold was set to filter features that showed a strong correlation with the personality values collected using the questionnaire. Notably, we used the *p*-value, rather than the r-value, for the following reasons: (1) because all correlation analyses were performed with the same sample size, the p- and r-values were monotonically correlated and (2) the *p*-value had strong statistical significance and was applicable to multiple channels. Moreover, we set the *p*-value as the basis of the nested models. Using elastic network regularization, the features that passed the screening were applied to build the regression model; the alpha parameter was empirically set at 0.75 (Zou & Hastie, 2005 [55]). The outer loop changed the *p*-value in the inner loop to select the optimal model. The *p*-value threshold traversed from 0.01 to 0.1 in increments of 0.01. For each set *p*-value threshold, the external loop was iterated N times, excluding the data of one participant each time; the data of the remaining N-1 participants were used as the training set to build the regression model (i.e., the internal loop), and this internal loop was applied to the participant left out to obtain the cross-validation results (predicted working age values). To assess the performance of each model, we calculated the Spearman correlation coefficients between the EEG-assessed and actual personalities of all participants. In the optimal model, the highest coefficient was 3000 cycles. The optimal models for the five personalities showed *p*-values of 0.02, 0.07, 0.08, 0.05, and 0.07, and the number of features selected for these models was 108, 78, 69, 53, and 128, respectively.

### 2.3.3. Evaluation of the EEG-Assessed Values

The assessment accuracy of the model was evaluated. We calculated the correlated indicators of the EEG-assessed and actual values for all participants and their subgroups. Next, we analyzed the external validity by asking all participants to answer whether the images were safe by pressing a button. To compare the accuracy of the assessment of these indicators, the responses and past casualty experiences were counted, and subsequently, the out-of-sample reliability was calculated. Prior to the model training, we withdrew data from eight participants to form a lockbox. Then, further validation was performed. We built a separate assessment model for the specific type of work hazard at a given height (ERP data from only 56 work-at-height trials were selected). The specific population with a lower education level (data from participants with junior high school or junior high school education and below) and the specific features of the classical ERP components were used to compare the assessment performance of the new model with that of the original model.

## 3. Results

### 3.1. Self-Assessed Personality Results

We recorded the personalities of the 61 workers using the Big Five questionnaire (Figure 3). There were no distinct differences in the Big Five traits between all participants and those with a low educational level. The results of the Big Five traits (agreeableness, conscientiousness, neuroticism, extraversion, and openness) were as follows: all participants: mean = 4.03, 4.10, 5.42, 5.52, and 4.29, respectively; standard deviation (SD) = 1.03, 1.06, 0.80, 1.00, and 1.20, respectively; participants with a low educational level: mean = 3.95, 4.06, 5.40, 3.51, and 4.21, respectively; SD=1.06, 0.65, 0.80, 1.01, and 1.18, respectively; $t(113)$ = 0.39, 0.28, 0.14, 0.05, and 0.33, respectively; $p$ = 0.70, 0.78, 0.89, 0.96, and 0.75, respectively.

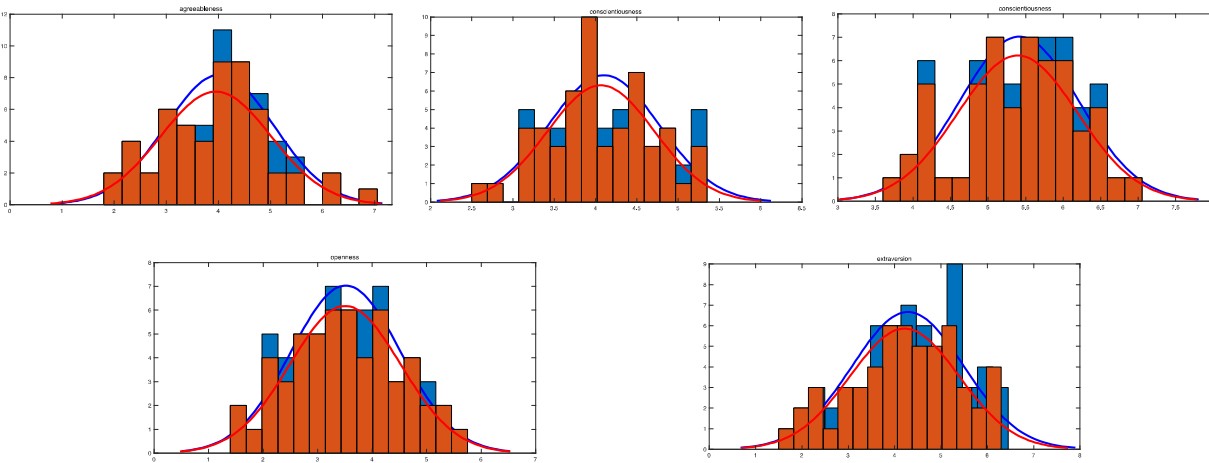

**Figure 3.** Distribution of the Big Five personality traits: All participants (blue) and those with a lower education level (red).

### 3.2. Behavioral Results

A total of 60 safe and 60 hazardous stimuli images were randomly presented to the participants who expressed their judgments by pressing a button. The hazard recognition process required the participants to retrieve the corresponding information from the images and make judgments. The average response time of the participants was 4.40 ± 1.28 s, and the average correct response rate was 61.81 ± 6.58%, indicating that the participants maintained their concentration throughout the experiment.

### 3.3. ERP Results

The total sample was divided into three groups according to the personality scores: high, medium, and low. The mean ERP images for all channels in Figure 4 show that the

significant ERP components caused by the experimental stimuli appeared at 120–180 (P1), 180–240 (N2), and 260–340 ms (P3).

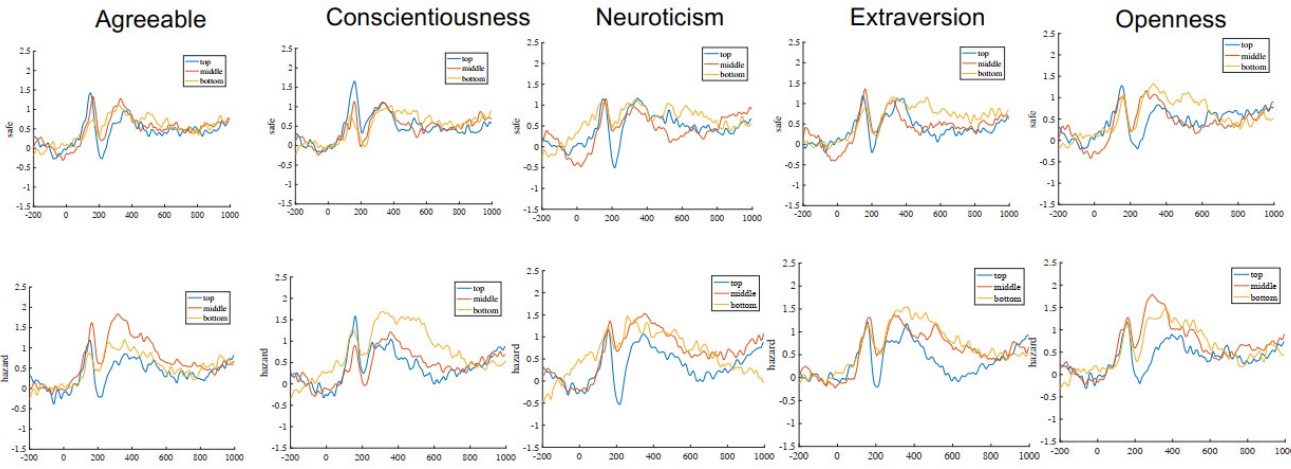

**Figure 4.** ERP image of the Big Five traits (mean of all channels).

We identified some components with significant correlations (Figure 5). With the safe stimuli, the P1 component in the parietal–occipital area of openness (r = 0.247, *p* < 0.1) showed a positive correlation with the actual value; the N2 component in the parietal–occipital area of agreeableness (r = −0.270, *p* < 0.05), neuroticism (r = −0.233, *p* < 0.10), and openness (r = −0.240, *p* < 0.10) showed a negative correlation with the actual value; and the P3 amplitude of the central area of agreeableness (r = 0.297, *p* < 0.05) showed a positive correlation with the actual value. Regarding the hazardous stimuli, the N2 amplitude in the parietal–occipital area of agreeableness (r = −0.284, *p* < 0.05), neuroticism (r = −0.227, *p* < 0.10), extraversion (r = −0.217, *p* < 0.10), and openness (r = −0.246, *p* < 0.10) showed a significant negative correlation with the actual value; meanwhile, the P3 amplitude in the parietal–occipital area of agreeableness (r = −0.342, *p* < 0.05) and openness (r = −0.297, *p* < 0.05) showed a significant negative correlation with the actual value.

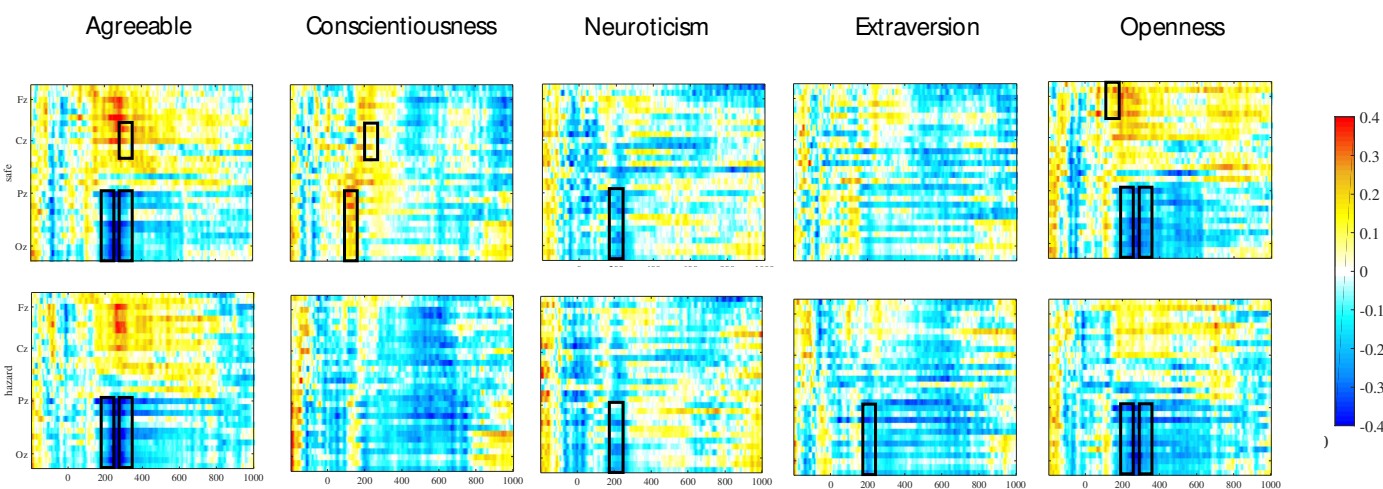

**Figure 5.** Correlation between the electroencephalography-assessed and self-reported Big Five traits. The black boxes indicate the significant parts.

*3.4. ERP-Based Personality Assessment Model*

To evaluate the validity of the model, we calculated the Spearman's correlation coefficients between the EEG-assessed and actual personality traits. The ERP features retained in the sparse regression model were filtered to include not only the time windows analyzed

above but also some features in the prestimulus and postprocessing phases (Figure 6). The EEG-assessed values for each personality trait showed a significant correlation with the actual values in the assessment model (Figure 7).

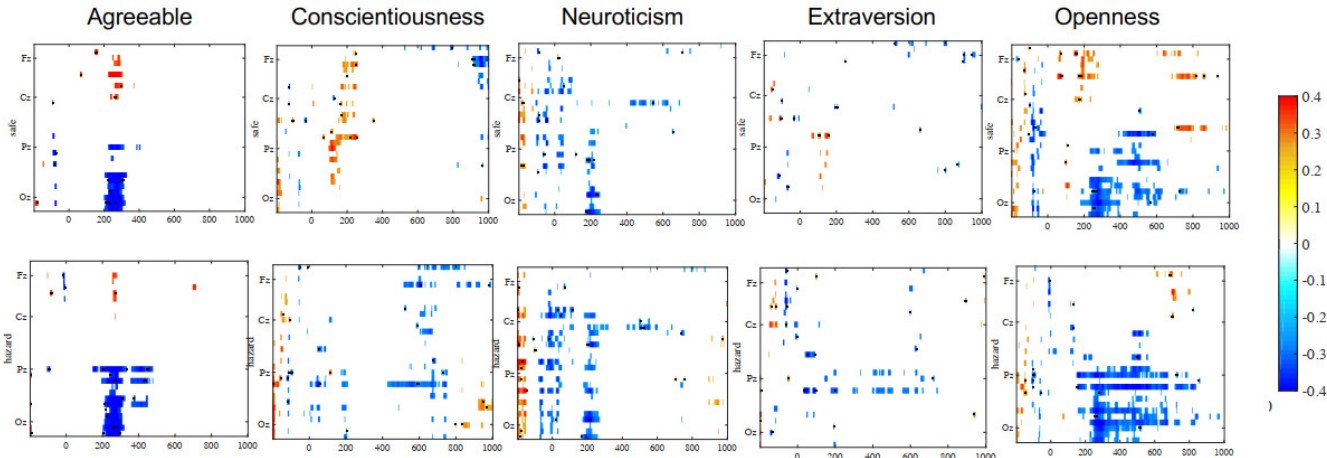

**Figure 6.** Screened event-related potential variables. The black dots indicate the final adopted variables.

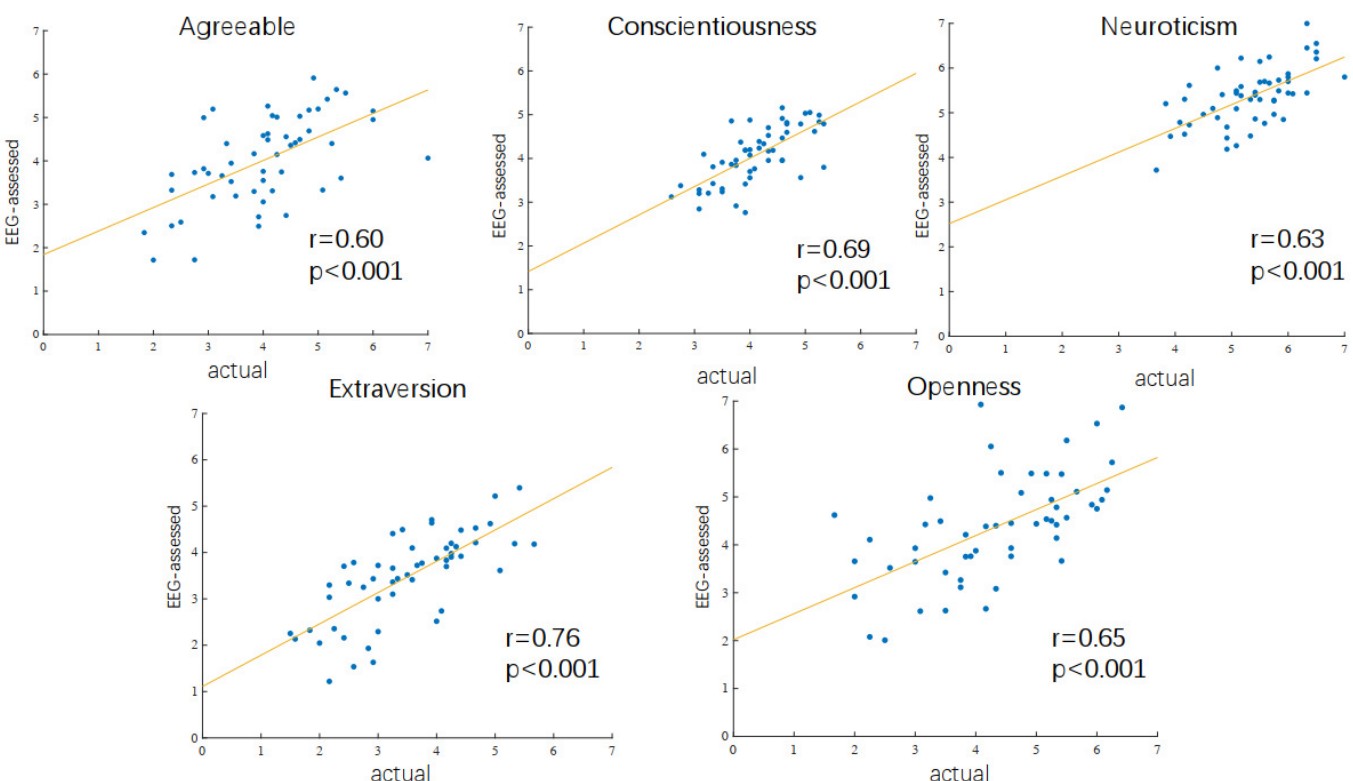

**Figure 7.** The Big Five traits of each participant: EEG-assessed and self-reported. EEG, electroencephalography.

The absolute error between the assessment and actual values was basically within 1.0 on the 7-point scale (Figure 8a), indicating that the model could assess personalities accurately.

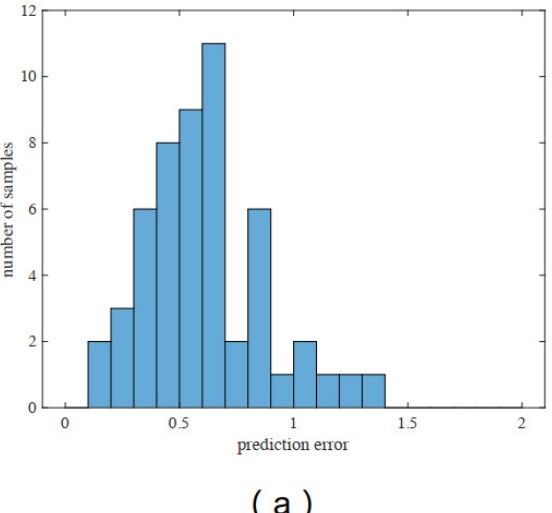
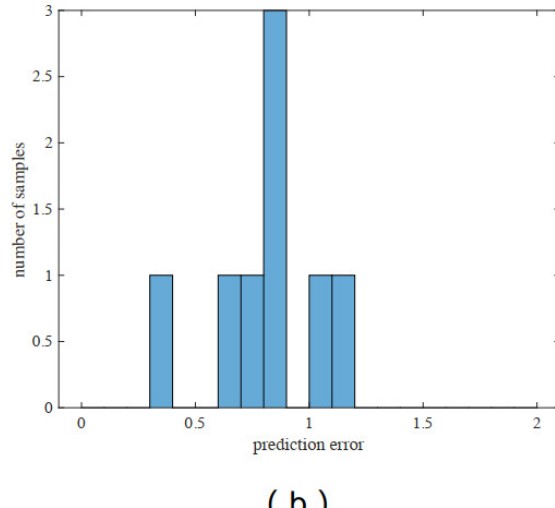

( a )                ( b )

**Figure 8.** The absolute error of the predicted value of the model. (**a**) All participants; (**b**) participants in the lockbox.

### 3.5. External Validity

We evaluated the behavioral performance of all participants and used it to assess the external validity of the model. All participants identified the images presented in the experiment as safe or hazardous and reported their casualty experience by filling out the questionnaire. We built regression models between the EEG-assessed and actual values of the three indicators. As shown in Figure 9, (1) the EEG-assessed values of conscientiousness, neuroticism, and extraversion were relatively close to the actual values in terms of goodness-of-fit; (2) the correlations between conscientiousness and all three indicators were high, indicating that conscientiousness may influence hazard recognition performance; and (3) extraversion also had a more significant correlation with the hit rate for hazards, indicating that extraversion has some effect on hazard recognition performance.

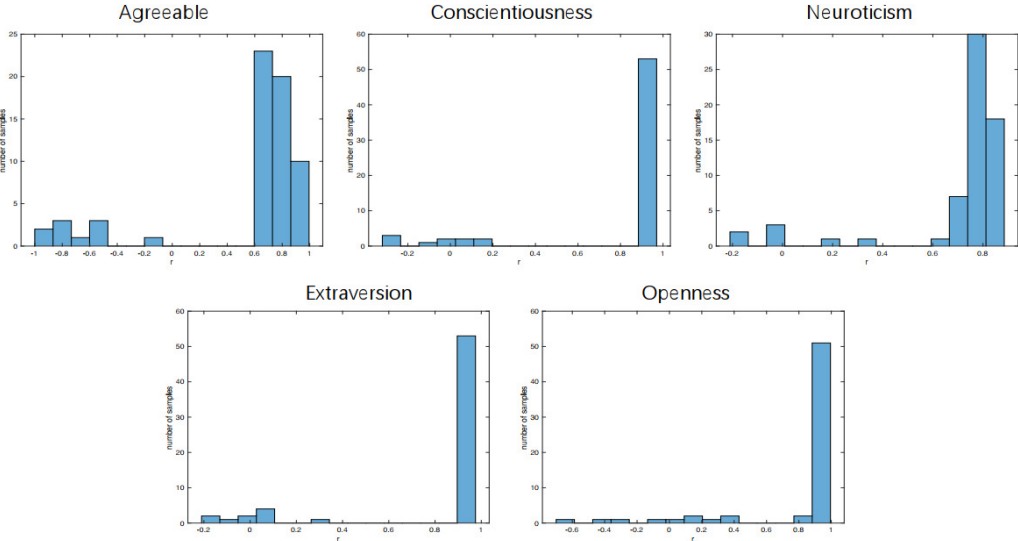

**Figure 9.** Histogram of the participant-wise correlations of the Big Five personality constructs between the electroencephalography-predicted and self-reported scores.

### 3.6. Out-of-Sample Reliability

Eight presampled participants were randomly selected from the database for validation. Overall, the analysis showed that the absolute errors between the EEG-assessed and

actual values were mainly within 1.0 on the 7-point scoring system (Figure 8b). The accuracy of the model in the lockbox was close to that of the normal sample.

### 3.7. Further Verification

The samples with high-altitude work and a low educational level were examined. The analysis revealed a high correlation between the EEG-assessed and actual values (Figure 10). This indicates that neither a specific group, such as those with a low educational level, nor a specific type of hazard, such as high-altitude work, interfered with the constructed model. Additionally, an attempt was made to assess the validity of this model by using typical ERP components to construct it. Specifically, the input features were set as the average amplitudes of the ERP components in the time window. The model was trained using the same machine-learning algorithm; however, the accuracy of the model was relatively low (Figure 11), demonstrating the necessity of including complete spatial and temporal ERP inputs.

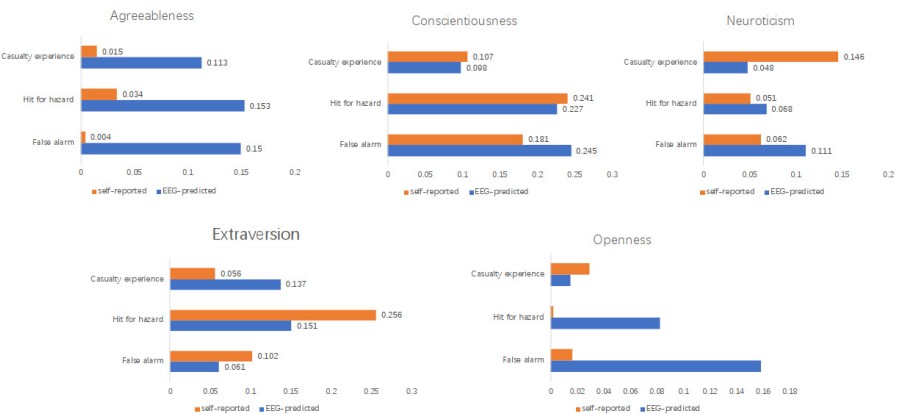

**Figure 10.** External validity between the EEG-assessed and self-reported Big Five traits. EEG, electroencephalography.

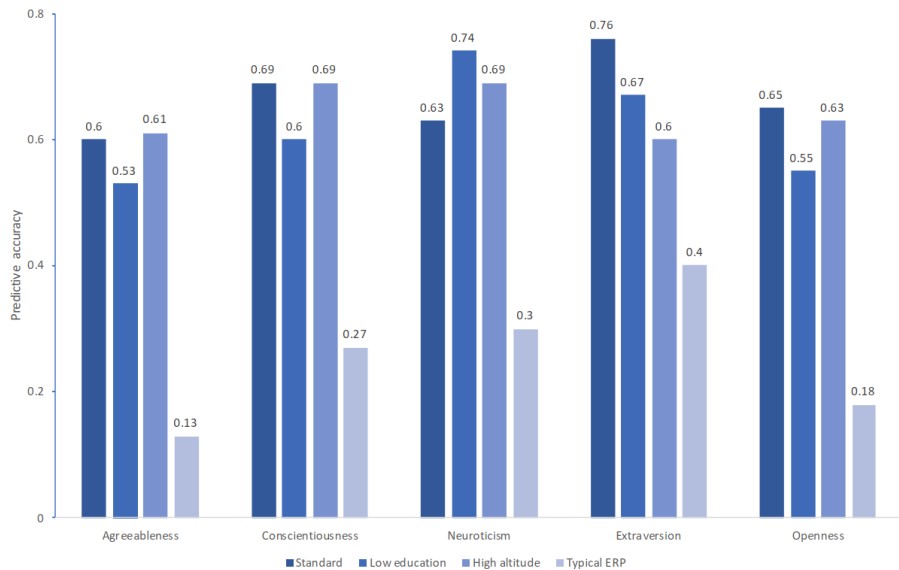

**Figure 11.** Influence of the input variables on the model predictive performance. ERP, event-related potential.

## 4. Discussion

In this study, we established an assessment model to evaluate workers' personalities by intuitively extracting physiological signals to replace the traditional questionnaire with the aim of avoiding falsification and reducing the HMC interface. The intuitive extraction of physiological signals using EEG was convenient, had a short processing time, and required

only simple experimental stimuli. In terms of the accuracy of the assessment, our model had a higher reliability and more comprehensive range of application scenarios than those in prior studies conducted with only intergroup model comparisons. It could automatically assess individual characteristics, thereby providing a basis for the input of human-related information into an HMC. The correlation analysis showed that the correlation coefficients between the EEG-assessed and actual values of each personality were all within 0.6–0.8, and the model performed better than the correlation values obtained from other EEG-related studies (Blankertz et al., 2010 [56]; Ihara et al., 2021 [57]). In the further validation analysis, the model was able to assess personality in different populations and under different hazard types, further illustrating its reliability and broad application.

In our study, the EEG-assessed conscientiousness values showed a significant correlation with the safe stimuli and a significant negative correlation with the hazardous stimuli. This is consistent with previous findings that conscientiousness and hazards are negatively correlated with each other (Beus et al., 2015 [1]; Eustace et al., 2018 [58]). Meanwhile, extraversion and neuroticism were correlated with the N2 component, which has been previously reported to be strongly associated with recognition tasks (Kopp et al., 2020 [59]). Pourmazaherian et al., (2018 [60]) stated that individuals with lower neuroticism would not focus on worrying when performing a task and would have better potential to concentrate on their tasks when compared with their counterparts. This is in line with the correlation between neuroticism and the N2 component derived in our study. Several studies have also suggested that extraverted individuals are generally good decision-makers because they actively seek and ask for more information to make decisions; accordingly, this makes them more aware of their situations (Staw & Barsade, 1993 [61]). This is consistent with previous findings that the N2 component reflects an individual's tendency to gather information regarding a task, and a greater N2 amplitude corresponds to a more substantial processing of the target information (Loughnane et al., 2016 [62]). However, these aforementioned correlations are relatively weak (0.217–0.342) and challenging to use in personality assessment. Contrastingly, machine-learning algorithms can process a large number of ERP features. In another study, we focused on the significant correlation between the N2 and P3 components and work experience. Notably, personality was correlated not only with the N2 component but also with work experience; similarly, personality correlated with the P3 component. This is consistent with previous findings that the P3 component is related to the degree of engagement in cognitive tasks (Kimura et al., 2008). The stronger correlations with the P3 component suggest that personality could be a better indicator of task performance than work experience.

While training the feature model with a single type of stimulus, the accuracy for the five personalities was 0.54, 0.64, 0.56, 0.69, and 0.5 for the safe stimuli and 0.51, 0.68, 0.38, 0.56, and 0.49 for the hazardous stimuli, respectively. The overall correlations obtained with only the safe or hazardous stimuli were both good but lower than those with both safe and hazardous stimuli. The models of conscientiousness under one and two stimuli were comparable, indicating that conscientiousness is better assessed from both the aspects of safety and hazard. Hasanzadeh et al., (2019 [8]) argued that workers with high conscientiousness are less likely to ignore hazards because they allocate more attention to identifying hazards on-site than their counterparts. Conversely, Wallace and Vodanovich (2003 [63]) showed that the failure to identify hazards was more likely to occur in situations with lower levels of conscientiousness than in those with higher levels. The model performed better for conscientiousness with hazardous stimuli, except for other personalities, which performed better with safe stimuli. This indicates that conscientiousness is more accurate when assessing based on identifying hazards rather than safety, implying that individuals with high conscientiousness tend to have better hazard recognition abilities than their counterparts.

Regarding external validity, the EEG-assessed and actual values of conscientiousness, neuroticism, and extraversion were closely related to the indicators of casualty experience, hit for hazard recognition, and false alarm, thereby reflecting the potential to assess practical hazard recognition ability. The EEG-assessed and actual values of conscientiousness

correlated with all performance indicators, indicating that conscientiousness has some advantages in assessing workers' hazard recognition ability. Combined with the importance of conscientiousness in the hazard recognition process mentioned earlier, we could consider developing worker conscientiousness in future safety training to improve hazard recognition performance. In addition, extraversion was correlated with the hits for hazard recognition, indicating that extraversion influences the workers' ability to accurately identify hazards. This may be related to the fact that extraverted individuals often request more information to increase their awareness of the situation when making decisions (Staw & Barsade, 1993 [61]), which makes them more likely to identify hazards. This is consistent with the personality traits. The effects of agreeableness and openness on hazards are primarily reflected in teamwork (Boyce & Wood, 2011 [64]; Mount et al., 1998 [19]) and conservatism in practices (Beaty et al., 2016 [65]; Matzler et al., 2006 [66]). Our experiment was modeled by the EEG recordings of workers' hazard recognition processes. The related process rarely implicated these aspects and therefore performed poorly in assessing these two personalities. In contrast, conscientiousness and extraversion were significantly associated with and influenced the workers' attention allocation and search strategies when exposed to hazards. Workers with higher commitments allocate their limited attentional resources appropriately to identify fall hazards in images (Hasanzadeh et al., 2019 [8]). Individuals with very high neuroticism may also become distracted because of anxiety and worry. These characteristics are closely related to hazard recognition.

This study had some limitations. First, the proposed assessment model was based on the EEG signals obtained during the hazard recognition process; therefore, the assessment ability was limited. For personalities that were less related to the hazard identification process in this experiment, the performance was not as good as that of other personalities. In the future, the EEG method should be applied to more hazard recognition processes to better assess personality traits. Second, the ERP features extracted from the EEG signals were used as the basis for model building. Other information in the EEG signals could be mined to study the worker differences. Third, although this study used machine-learning methods and built a lockbox for research, the experimental data were all from the same experiment; test–retest experiments need to be conducted in the future to verify the reliability of the model. Additionally, gaps remained between the models constructed in this study and their practical application. For example, the experiment used static images as stimuli that could not simulate the dynamics of an actual construction site. In the future, VR scenes could be used as experimental stimuli to obtain EEG signals closer to that which would be obtained from the actual construction site. Furthermore, personality monitoring during a task has high requirements for both acquisition and computing equipment, and hardware matching and artifacts should be fully considered in future practical applications.

Overall, we increased the analytical dimensions and compared them with the results of our previous study (e.g., work experience) (Wang et al., 2022 [53]) by using an analytical method to correlate various personality dimensions with EEG components and task performance. Thereafter, we compared these correlations across personality dimensions, identifying salient individual characteristics (conscientiousness, neuroticism, and extraversion) that indicate task performance. Our results indicate that future research can use the proposed analytical approach to measure multidimensional dispositional factors and future intuitive HMC technologies could capture various combinations of components and assess different dispositional factors simultaneously.

## 5. Conclusions

In this study, we used a machine-learning approach to process the EEG signals of workers during image observation to build a model for assessing their scores for the Big Five traits. The assessment results derived from the proposed model were correlated with the scale results. The external validity confirmed that the model had a good assessment performance for the Big Five traits. Theoretically, this study demonstrated the feasibility of assessing worker characteristics using their EEG signals during hazard recognition. In the future, this method could be

applied to assess various individual characteristics and provide a research basis for individual differences. Moreover, we solved the risk of overfitting caused by the high-dimensional features of EEG signals. Cross-validation, namely, Leave-One-Out Cross Validation and lockbox validation, was performed to ensure the reliability of the model. Furthermore, we constructed both inner and outer loops. The inner loop reduced the dimensionality of the high-dimensional features by building a sparse regression model and filtered the features by setting a correlation gradient with a *p*-value threshold. The outer loop contained a leave-one-out validation algorithm that sought the optimal regression model by changing the *p*-value threshold set in the inner loop. The established dual-validation model can provide a basis for future research.

In practice, the personality assessment model established herein could provide a parametric basis for intelligent devices in HMC. Workers' personality information can then be obtained directly by monitoring their EEG signals during tasks, without the need for prior investigation and input of personality parameters, which can reduce the HMC interface. Simultaneously, the model can allow intelligent devices in HMC to provide idiosyncratic and effective information to workers, improve hazard recognition performance, and contribute to the development of intuitive HMC technologies. The proposed model promotes an automated personality assessment during hazard recognition tasks and is more efficient than a self-reported approach. The assessment results can serve as a reference for targeted worker retraining and task allocation to improve performance in construction safety.

**Author Contributions:** Methodology, M.W.; Software, M.W.; Validation, M.W.; Formal analysis, M.W.; Investigation, P.-C.L.; Resources, P.-C.L.; Data curation, M.W.; Writing—original draft, M.W.; Writing—review & editing, P.-C.L.; Visualization, M.W.; Supervision, P.-C.L.; Project administration, P.-C.L.; Funding acquisition, P.-C.L. All authors have read and agreed to the published version of the manuscript.

**Funding:** This study was supported by the National Natural Science Foundation of China [grant number 51878382].

**Data Availability Statement:** All data, models, and codes that support the findings of this study are available from the corresponding author upon reasonable request (items: EEG data, database containing the image stimuli, and MATLAB code for developing the assessment model).

**Conflicts of Interest:** The authors declare no conflict of interest.

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
