# Peer review of "Personality Assessment Based on Electroencephalography Signals during Hazard Recognition"

_sustainability, doi:10.3390/su15118906_

Round 1

Reviewer 1 Report

The author needs to update the related work with state-of-the-art studies.
Give more explanations regarding figures.

 The research findings and contribution need to be stated clearly. As well as the obtained results in this paper.

Check the paper language and make sure all language errors have been fixed.

Please use the complete form of abbreviations in the abstract.

The abstract also should be revised according to the main idea of this research and the main motivation behind the proposed research.

 The research findings and contribution need to be stated clearly. As well as the obtained results in this paper.

Cite below paper 

Achanta, Sampath Dakshina Murthy, Thangavel Karthikeyan, and R. Vinoth Kanna. "Wearable sensor based acoustic gait analysis using phase transition-based optimization algorithm on IoT." International Journal of Speech Technology (2021): 1-11.https://doi.org/10.1007/s10772-021-09893-1

Sampath Dakshina Murthy, Achanta, Thangavel Karthikeyan, and R. Vinoth Kanna. "Gait-based person fall prediction using deep learning approach." Soft Computing (2021): 1-9. https://doi.org/10.1007/s00500-021-06125-1

Reviewer 2 Report

This paper  proposed an implicit electroencephalography (EEG)-based measure to automatically assess worker personalities, underpinning the development of HMC techniques. It is meaningful and fits the scope of Sustainability well. It wrote quit well. In my opinion, it is accepted though minor English expression and grammar polishing.

Reviewer 3 Report

The manuscript deals with the use of ECG for inferring the personality traits of workers when facing with hazardous/safe situations. The topic is interesting for the Sustainability community and the paper is well-written and well-organised.

However, two issues do not make the paper suitable for publication in its current form. My remarks are reported below.

1. This remark refers to the methodological approach.

The novelty of your contribution is to assess the personality traits of workers using only ECG but how ECG outputs are linked with the personality traits is not clear. Tht is to say, as the personality trait self-assessment can be distorted by distorted self-judgments, the said relationship (ECG-traits) can be distorted as well. In other words, an unsupervised clustering approach can cluster ECG outputs, but labelling clusters with personality traits can be pretentious. Otherwise, if a supervised classifier is used, the examples must be labelled properly for training the classifier properly. I guess that you used the latter approach but this must be clarified. Hence, in order not to confuse the reader, I suggest clarifying the adopted approach from the beginning.

2. The machine learning approach has been neglected. I suggest enriching this part as well.

Best regards
